# HANDLING UNSTRUCTURED DATA FOR OPERATOR LEARNING USING IMPLICIT NEURAL REPRESENTATIONS

**Thomas X Wang & Patrick Gallinari**
ISIR, Sorbonne Université
Paris, France
`thomas.wang@sorbonne-universite.fr`

## ABSTRACT

Operator learning methods are too often constrained by a fixed sampling of both the input and output functions. We propose a novel method to allow current operator learning methods to learn on any sampling. We show that our method can perform inference on unseen samplings, and that it allows returning outputs as continuous functions.

## 1 INTRODUCTION

An operator is a mapping between two function spaces, an example of operator is the mapping between the initial condition of a Partial Differential Equation (PDE), and the solution of the PDE at a later time. PDEs are ubiquitous in physics, where they model complex systems in many fields (fluid dynamics, electrodynamics...). They can be analytically intractable and require the use of very computationally-expensive numerical solvers. Recently, Deep Learning has emerged as a potential technique for helping to solve PDEs faster.

Recently, two operator learning approaches have been proposed: FNO (Li et al., 2021) and Deep-ONet (Lu et al., 2021), which have been followed by many applications. However, they come with some limitations over the structure of the input data: DeepONet requires the same sampling across all its input functions, while FNO requires the functions to be sampled on regular grids to be able to perform the Fast Fourier Transform. Such sampling is not always obtainable in real-life, for example in the case of drifting buoys to measure ocean currents. Implicit Neural Representations (INRs) are neural networks that take coordinates as inputs. They allow the representation of different types of signals such as images, 3D shapes via a continuous neural network by using sinusoidal activation functions (Tancik et al. (2020), Sitzmann et al. (2020)), while capturing the high-frequency details of the signal. They have exhibited interesting properties for data compression (Chen et al., 2022), video generation(Yu et al., 2022), and physics-learning(Yin et al., 2023).

In this work, we extend current operator learning algorithms by augmenting the available data to fit the algorithm requirements, using INRs. We show that our approach outperforms conventional interpolation methods.

## 2 METHODOLOGY

**Problem Setting**  We aim to learn an operator $\mathcal{F} : f \mapsto g$, where $f$ and $g$ are two functions defined on spatial domains $\mathcal{D}$ and $\mathcal{D}'$. In practice, we only have access to the values of these functions where they are measured. Thus, our dataset is composed of couples of functions $(f_i, g_i)_{i=1..n}$, evaluated at the respective sets of locations $\mathcal{X}_i = (x_i^j)_{j=1..p_i}$ and $\mathcal{Y}_i = (y_i^j)_{j=1..q_i}$, where $x_i^j \in \mathcal{D}$ and $y_i^j \in \mathcal{D}'$. We can rewrite it as learning a mapping between the sets $(f_i|_{\mathcal{X}_i})_i$ and $(g_i|_{\mathcal{Y}_i})_i$. We note that the number and positions of the sampled points can vary across each considered couple $(f_i, g_i)$. Additionally, note that even if the domains $\mathcal{D}$ and $\mathcal{D}'$ are the same, the sensor locations of the input functions and of the output functions can be different.

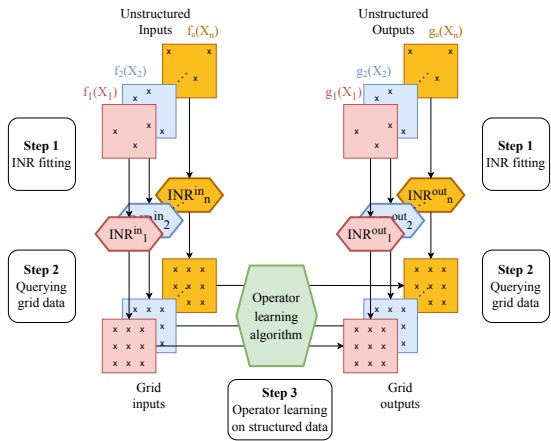

Figure 1: Three-step training architecture

**Architecture** Our method consists of three steps, which are depicted in Fig. 1. ● First, we approximate the unknown input and output functions $f_i$ and $g_i$ using Implicit Neural Representations: for each $f_i$ function, we train an INR $INR_i^{in}$ to fit $f_i$ at the locations of $\mathcal{X}_i$. As in Dupont et al. (2022), we use modulated INRs: the networks share common parameters, which are then modulated differently for each INR, so that each INR also benefits from the common information in the data. The same process is done for the output functions $g_i$. ● In a second time, we use these functions $INR_i^{in}$ and $INR_i^{out}$, respectively approximating $f_i$ and $g_i$, to provide values on regular grids $\mathcal{X}_{grid}$ and $\mathcal{Y}_{grid}$ by querying these INRs on the corresponding grids. ● Finally, we train our operator learning algorithm on the surrogate regular-grid data: $(INR_i^{in}(\mathcal{X}_{grid}), INR_i^{out}(\mathcal{Y}_{grid}))_i$.

## 3 EXPERIMENTS

We demonstrate our method on 2 different datasets: the Shallow-water equation, and the Navier-Stokes equation. We want to learn the operator $\mathcal{F}$ which maps an initial condition $u_0(x) = u(x, t = 0)$ to its state at a later time: $u_1(x) = u(x, t = 1)$. We use FNO as the operator learning algorithm, but our method is applicable to other methods that are constrained by the sampling of its input data.

The input data is measured at different randomly-sampled sensor locations for each initial condition. The sampling rate $s$ is the ratio of the available sensors, compared to a regular grid of reference. We compare our method for handling unstructured data with more traditional interpolation methods: linear and cubic interpolations. We show in Table 1 that our augmentation with INRs outperforms other unbiased interpolation methods.

Table 1: Test relative MSE ($\downarrow$) on datasets sampled with rates $s$ from $0.05$ to $0.9$.

|  | Shallow-water | | | | Navier-Stokes | | | |
|---|---|---|---|---|---|---|---|---|
|  | $s = 0.05$ | 0.15 | 0.5 | 0.9 | 0.05 | 0.15 | 0.5 | 0.9 |
| Linear | 2.58e-3 | 1.41e-3 | 5.62e-4 | 1.89e-4 | 1.47e-2 | 5.58e-3 | 1.44e-3 | **8.74e-4** |
| Cubic | 2.40e-3 | 1.13e-3 | 4.54e-4 | 1.63e-4 | 1.00e-2 | 2.30e-3 | 1.08e-3 | 9.45e-4 |
| Ours | **3.09e-4** | **2.40e-4** | **1.57e-4** | **1.27e-4** | **1.17e-3** | **1.28e-3** | **9.81e-4** | 1.26e-3 |

## 4 CONCLUSION

We show that our approach can extend operating learning methods for unstructured data using INRs, and shows clear advantages with respect to standard data augmentation techniques.

URM STATEMENT

Author Thomas X Wang meets the URM criteria of ICLR 2023 Tiny Papers Track.

ACKNOWLEDGEMENT

We acknowledge financial support from DL4CLIM (ANR-19-CHIA-0018-01) and DEEPNUM (ANR-21-CE23-0017-02) ANR projects.

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

## A  RELATED WORK

**Implicit neural representations**  (INRs) are neural networks which are used to represent signals continuously, as opposed to traditional discrete methods. They learn a mapping between coordinate $x$ and the value $\Phi(x)$ of the signal at the specified coordinate $x$. By using sinusoidal activation functions instead of more traditional activations, Fourier features (Tancik et al., 2020) and SIREN (Sitzmann et al., 2020) have shown to be able to represent high-frequency signals. The SIREN architecture is as follows :

$$\Phi\left(\mathbf{x}\right) = \mathbf{W}_n\left(\phi_{n-1}\circ\phi_{n-2}\circ\ldots\circ\phi_0\right)\left(\mathbf{x}\right)+\mathbf{b}_n, \quad \mathbf{x}_i \mapsto \phi_i\left(\mathbf{x}_i\right)=\sin\left(\mathbf{W}_i\mathbf{x}_i+\mathbf{b}_i\right). \quad (1)$$

While a SIREN network can only represent one signal at a time, modulated INRs (Dupont et al., 2022) allow for representing a whole dataset of signals: while sharing a common core INR, they modulate each of this core INR's layer $i$ differently for each signal $j$, as in equation 2. The modulations $\gamma_j$ and $\beta_j$ are obtained via linear transformations from a learned embedding $z_j$.

$$\phi_i^j : \mathbf{x}_i \mapsto \phi_i\left(\mathbf{x}_i\right)=\sin\left(\gamma_j * \mathbf{W}_i\mathbf{x}_i+\mathbf{b}_i+\beta_j\right). \quad (2)$$

## B  RESULTS VISUALIZATION

We use the incompressible 2D Navier-Stokes equation dataset from Li et al. (2021), and the 2D Shallow-water dataset from Takamoto et al. (2022). Results from the Shallow-water dataset are shown in figure 2.

## C  IMPLEMENTATION DETAILS

The model was implemented using PyTorch (Paszke et al., 2019), as well as the Hydra (Yadan (2019)) and einops (Rogozhnikov, 2022) libraries. The operator learning algorithm was implemented using the FNO algorithm (Li et al., 2021).

In our case, the embbeddings $z_j$ used to obtain the INR modulations are learned via meta-learning, as in Serrano et al. (2023).

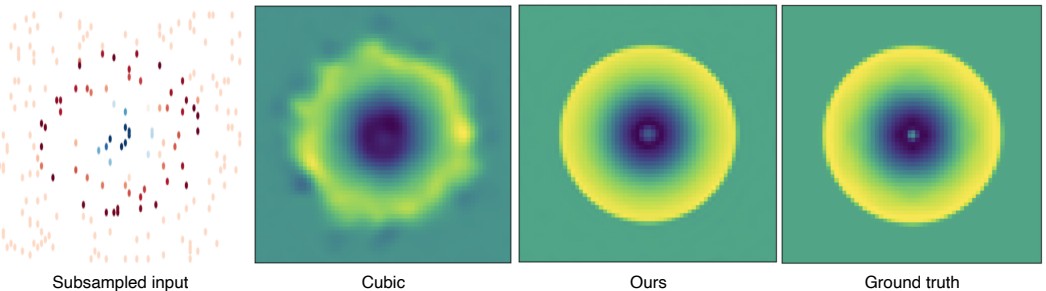

Figure 2: Shallow water comparison with $s = 0.05$. The subsampled input is $u_0(x)|_{\mathcal{X}_i}$. The second and third images show predictions of the target function $u_1(x)$ obtained via cubic interpolation, and our method. We can see that our method provides much better results, and is very close to the groundtruth function.

