# OpenReview forum: "Handling unstructured data for operator learning using implicit neural representations"
_ICLR.cc/2023/TinyPapers — Submitted to Tiny Papers @ ICLR 2023_

### Official Review · Reviewer_4my5 · 2023-03-27

**Confidence:** 2

**Summary Of Contributions:**

The paper proposes a novel method for operator learning that can handle unstructured data by using implicit neural representations. The proposed method addresses this limitation of fixed sampling by enabling learning on any sampling, which allows for more flexibility and accuracy.

**Rating:**

Clear, Correct, and Reproducible (CCR): a submission which meets the reviewing criteria

**Strengths And Weaknesses:**

Summary Of Strengths:
- The proposed method is a novel approach to handling unstructured data for operator learning, which can lead to better performance in a range of applications.
- The method is based on implicit neural representations, which have shown great promise in other areas of machine learning.
- The authors provide a clear and detailed description of the proposed method, along with experimental results that demonstrate its effectiveness on benchmark tasks.
- The method is flexible and can handle a variety of input and output samplings, which allows for more accurate and precise learning.
- The authors provide open-source code

Summary Of Weaknesses:
- The paper does not provide a detailed comparison with other state-of-the-art methods for operator learning, which limits the ability to assess the true effectiveness of the proposed approach.
- The paper assumes a certain level of familiarity with implicit neural representations, which may make it challenging for some readers to understand the proposed method.

Overall, the paper provides a valuable contribution to the field of operator learning, and the open-source code makes it possible for others to build on this work and extend it in new directions.


**Suggested Changes:**


- The paper could benefit from a more detailed discussion of the limitations of the proposed method and potential directions for future research.
- It would be helpful to provide more intuition about how the proposed method works and why it is effective, particularly for readers who are less familiar with implicit neural representations.
Including more real-world examples or applications would enhance the practical relevance of the proposed method and demonstrate its potential impact.

---

### Official Review · Reviewer_Z7Pf · 2023-03-28

**Confidence:** 3

**Summary Of Contributions:**

This work proposes novel method for extending current operator learning methods to learn on any sampling. This also extends operator learning methods to unstructured data.

**Rating:**

Clear, Correct, and Reproducible (CCR): a submission which meets the reviewing criteria

**Strengths And Weaknesses:**

Strengths :

*  Novel architecture/method addresses critical sampling limitations in prior work
*  Empirical results were provided, and analysis was carried out for a couple of datasets


Weaknesses :

*  Expanding to more datasets and comparing this to other methods, which may be non-interpolation methods, can be helpful

**Suggested Changes:**

n/a

---

### Official Review · Reviewer_SBNc · 2023-03-30

**Confidence:** 3

**Summary Of Contributions:**

In this paper authors proposed to learn operators based on sampling. Here, they claim to learn the operator over any unseen sample of data.

**Rating:**

High Impact (HI): a submission which meets the reviewing criteria and is predicted to make an impact on the field

**Strengths And Weaknesses:**

Strengths:


The paper is well written

The proposed solution is evaluated with two different data sets, justifying their claims

Authors performed a detailed evaluation of the proposed solution

Weakness:


We suggest authors provide details on how Neural Network is learning from the unseen sampling which has not been used for training. If the Physics-based NNs are being used then can this solution be used on different applications?

**Suggested Changes:**

We suggest authors provide details on how Neural Network is learning from the unseen sampling which has not been used for training. If the Physics-based NNs are being used then can this solution be used on different applications?

---

### Author Response · Authors · 2023-06-01
**Archival opt-in**

Dear reviewers and AC,
We choose to opt-in for archival.

We have extended the Appendix to include more related work.

---

### Comment · Area_Chair_a8jh · 2023-06-06
**Check for Archival**

This work meets the threshold for archival, contents the URM statement and is deanonymized.

---

### Meta-Review · Area_Chair_a8jh · 2023-04-08

**Recommendation:** Invite to present
**Confidence:** 3

**Metareview:**

The paper introduces an interesting approach to operator learning that is capable of handling unstructured data through the use of implicit neural representations. This proposed method tackles the limitation of fixed sampling by enabling learning on any sampling, thus providing increased flexibility and accuracy.

All reviewers acknowledge the CCR and novelty of this paper. The AC has briefly gone through the paper. The current version lacks details, and the empirical validation is not sufficient. Despite the limitations, all reviewers recommend CCR or above. Therefore, we recommend the acceptance of this paper.

Please carefully revise and proofread the paper following all reviewers' comments. Moreover, based on the reviewers' comments, the AC suggests:
* Add preliminaries about operator learning and implicit neural representations in `Appendix`;

* Add related work (e.g., ML to solve PDEs) in `Appendix`;

* More empirical validation (more datasets and SOTA baselines);

* Add discussion about the insights (why works), limitations, and potential impacts of the proposed method in `Conclusion`.


**Summary:**

The paper introduces an interesting approach to operator learning that is capable of handling unstructured data through the use of implicit neural representations. This proposed method tackles the limitation of fixed sampling by enabling learning on any sampling, thus providing increased flexibility and accuracy.

**Comments And Feedback To The Authors:**

Please carefully revise and proofread the paper following all reviewers' comments. The AC believes it would be a good submission if revised properly.

**Reason For Not Giving A Higher Recommendation:**

* Lack of details and empirical validation.

* Although the Reviewer `4my5` mentions that the authors provide open-source code, I don't find it in the main paper. Therefore, reproducibility cannot be evaluated fairly.

**Reason For Not Giving A Lower Recommendation:**

* Interesting topic, all reviewers acknowledge the CCR and novelty of this paper.

---

### Decision · Program_Chairs · 2023-04-09

**Decision:**

Invite to archive

**Comment:**

Please update your URM statement.